# A Simple Data-Parameters Balancing Framework for Early Ventricular Activation Origin Localization

## Abstract

Accurately identifying the site of origin (SoO) of early ventricular activation is crucial for catheter ablation, an effective therapeutic option for treating ventricular arrhythmia. However, due to the limited availability of clinical data and the errors introduced during data preprocessing, achieving precise localization remains a challenge. While deep learning models offer an end-to-end approach for data input in the ECG field, they often suffer from overfitting caused by limited training data, hindering continuous performance improvement. This paper proposes a **Sim**ple data-parameters **B**alancing framework for early ventricular activation **O**rigin **L**ocalization (SimBOL). By using onset-based data augmentation, the SimBOL method expands the training data derived from clinical samples. The framework utilizes a small-scale 1D convolution model that balances the relationship between available training data and model complexity, effectively mitigating overfitting and eliminating the need for extensive data preprocessing. SimBOL achieves a localization error as low as 9.83 mm, which meets clinical acceptable localization error ¡ 10 mm and outperforming existing methods in predicting the SoO of early ventricular activation. The discussion about data augmentation and model architecture on ECG signal processing, offering new insights into optimizing deep learning applications for ECG-based tasks.

## 1 Introduction

Catheter ablation has emerged as a useful therapeutic option for treating ventricular arrhythmias Levine et al. (2016), including ventricular tachycardia (VTs) and premature ventricular complexes (PVCs) Sapp et al. (2016); Cronin et al. (2019), both of which are major causes of sudden cardiac arrest (SCA). Accurately identifying the site of origin (SoO) of early ventricular activation is critical for electrophysiologists to focus their mapping and targeting efforts on a specific region during catheter ablation Asirvatham & Stevenson (2016). Currently, pace-mapping is one of the primary clinical mapping approaches for localizing the SoO Josephson et al. (1982). As illustrated in Figure 1, this method involves template-matching analysis and a "trial and error" approach, where intracardiac pace-mapping ECG morphologies (12-lead pacing ECGs) are compared to those of the targeted VT/PVC 12-lead ECGs using commercial modules Guenancia et al. (2022); Zhou et al. (2020b; 2021). Despite their utility, these systems do not provide direct localization information, pace-mapping is time-consuming and heavily dependent on the operator's expertise Zhou et al. (2023).

With the recent advancement of artificial intelligence (AI) in medicine Singhal et al. (2023); Lakkaraju et al. (2022); Alowais et al. (2023), several studies Yang et al. (2017); Pereira et al. (2019); Gyawali et al. (2019); Missel et al. (2020); Nakamura et al. (2021); Li et al. (2021); Chang et al. (2022) have explored the use of deep learning methods to identify the SoO of early ventricular activation, offering end-to-end signal processing and automatic extraction of complex patterns in ECG signals. In studies such as Gyawali et al. (2017; 2019); Missel et al. (2020), the real pacing-site data (ECG and corresponding pacing-site coordinates) available is relatively small (in the thousands) compared to the vast datasets used in deep learning (often in the millions or billions). The over-parameterization characteristic of deep learning models means that their effectiveness heavily depends on the number of training parameters and the size of the training data Allen-Zhu et al. (2019). When the number of training parameters is comparable to or far exceeds the available training data,

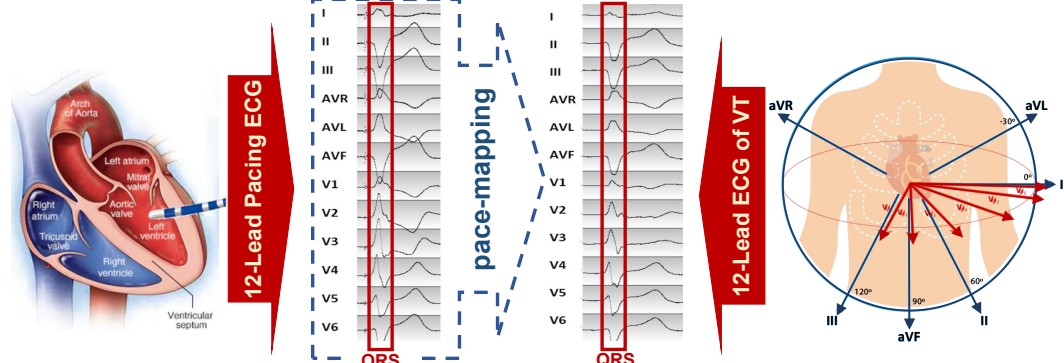

Figure 1: A schematic of the entire pace-mapping process is shown. On the left, the doctor uses electrodes to pace different areas of the ventricle, generating 12-lead pacing ECG. On the right, 12-lead ECG are collected from the body surface when an abnormality occurs in a specific heart region. The red solid line highlights the QRS complex of the ECG signals.

the risk of overfitting increases, limiting the model's ability to generalize and improve performance Dieterich (1995); Rice et al. (2020); Santos & Papa (2022). To address overfitting in deep models for this task, some studies Hendrycks et al. (2019); Jaiswal et al. (2020); Krishnan et al. (2022) have proposed pre-training with a generation task as a proxy, providing a warm start for subsequent model fine-tuning. Although these approaches can reduce overfitting, commonly-used architectures such as LSTM Graves & Graves (2012); Yu et al. (2019) and transformer Vaswani (2017); Devlin (2018) remain complex and cumbersome, adding additional challenges to the training process.

In this work, we propose a **Sim**ple data-parameters **B**alancing framework for ventricular arrhythmia **O**rigin **L**ocalization, (**SimBOL**). Not only does SimBOL outperform previous methods in terms of SoO localization accuracy, but it is also simpler, requiring neither specialized input data preprocessing Zhou et al. (2019) nor a pre-training process Gyawali et al. (2019). Due to the limited availability of the pacing-site data (ECG and corresponding site coordinates), SimBOL produces an onset-based data augmentation strategy. This approach expands the training dataset by randomly resampling within specific regions determined by onset points in ECG signals. Unlike previous models Yang et al. (2017); Pereira et al. (2019); Gyawali et al. (2019); Nakamura et al. (2021); Li et al. (2021); Chang et al. (2022); Missel et al. (2020), where the number of model parameters greatly exceeded the available training data, SimBOL provides a small-scale model that maintains a balance between data and parameters. It combines 1D convolution and fully connected layer, exploiting the robustness of 1D convolution in extracting features from time-series signals to handle augmented, irregular data effectively during training. The balance significantly reduces the model's overfitting and stabilizes its localization performance. Furthermore, SimBOL predicts site coordinates to calculate distance loss instead of traditional label prediction loss, reducing the potential biases introduced by operations like SoftMax in model training. Experimental results demonstrate that SimBOL achieves the SoO localization within 10mm, significantly better than the current most accurate method, SVR Zhou et al. (2019), by over 2mm. The discussion about data augmentation and model architecture on ECG signal processing, offer- ing new insights into optimizing deep learning applications for ECG-based tasks. In summary, the contributions of this paper are summarized as follows:

- This work introduces Onset-based data augmentation, a method that leverages physical characteristics of ECG signal. By resampling specific regions of the ECG data, this method effectively increases the amount of training data for deep learning models, providing critical data support for their application in the ECG field.

- This work proposes a data feature extraction model specifically designed for ECG signals. By appropriately stacking 1D convolution and fully connected layers, the model effectively extracts features from the augmented, irregular ECG signals. Its small scale offers a new direction to mitigating overfitting during model training in the ECG field.

- The SimBOL training framework achieves a coordinate prediction distance error of less than 10mm, significantly improving upon the current best localization method, which has an error of 12mm. Its minimal requirements for input data processing not only increase accuracy but also enhance convenience and efficiency in clinical settings.

## 2 RELATED WORK

### 2.1 EARLY VENTRICULAR ACTIVATION ORIGIN LOCALIZATION

Using 12-lead ECGs data to guide clinicians in locating the SoO of early ventricular activation has long been a focus of clinical and biomedical research. In 2017, Sapp et al. (2017) proposed subdividing the 16 segments of a generic left ventricle (LV) into 238 triangles. They collected a pacing-site database that included pacing 12-lead ECGs and corresponding pacing locations. By applying a multiple linear regression (MLR) method Zhou et al. (2019), they derived coefficients linking the pacing ECGs to their corresponding pacing locations. These coefficients were then used to predict the SoO of early LV activation when having a pacing ECG or VT/PVC ECG. In this approach, the ECG for each lead was represented by a single value based on the 120 ms QRS integral. However, this method is highly sensitive to the selection of the QRS onset and by focusing only on the integral of the first 120 ms of the QRS complex, it may provide a limited understanding of ventricular contraction, without fully capturing the dynamics of ventricular relaxation.

Due to AI's powerful data analysis and feature extraction capabilities, some studies Lai et al. (2021); Haq et al. (2021); Liu et al. (2023) are exploring the use of deep learning (DL) to process clinically relevant but "hidden" patterns in ECG signals, thereby assisting in identifying the SoO of early ventricular activation. Yang et al. (2017) employed a comprehensive DL approach using two CNNs (Segment CNN and Epi-Endo CNN) to localize PVC origin onto 25 specific ventricular segments (17 LV segments, 8 RV segments) and distinguish between endocardial and epicardial activation origins. Pereira et al. (2019) employed a shallow neural network for PVC detection, focusing on distinguishing between left and right ventricles. Tested on three datasets with 328 patients, the network demonstrated high specificity but relatively low sensitivity. Gyawali et al. (2019); Missel et al. (2020) used a generation task with GRU models, followed by fine-tuning for localization. The model achieved a mean localization accuracy of 12.84 mm within ten segments of a generic LV endocardium. Li et al. (2021) developed an optimized ResNet-18 network for localizing PVC origins. Chang et al. (2022) developed a DL model with two sets of CNN layers for predicting the Ventricular Arrhythmia origin. However, the performance of these models is significantly constrained by overfitting due to the limited availability of clinical data and the over-parameterized of deep models.

### 2.2 DIFFERENCE BETWEEN ECG AND SPEECH SIGNALS

Many speech-related tasks Dureja & Gautam (2015); Wali et al. (2022) use the spectral information of speech signals as input for model training. This is mainly because the content of human speech is primarily conveyed through frequency information, while amplitude and timbre only represent the loudness and quality of the sound Fitch (2000). In contrast, ECG data, being electrical signals, presents unique challenges; its amplitude, as a crucial component that carries important information about the strength of heart contractions Malmivuo & Plonsey (1995). Additionally, the frequency in ECG data reflects the distinct beating patterns of the heart's chambers. Furthermore, since heart signals are influenced by the body when transmitted to the body surface, 12-lead ECG data often suffers from significant noise interference Clifford et al. (2006), making it more difficult to capture heart rate frequency information accurately. As a result, directly applying speech-domain models to ECG data processing is not ineffective.

## 3 PACING-SITE DATA

### 3.1 PACING-SITE ECGS

Figure 2 presents a complete 12-lead pacing ECG collected in real time during clinical procedures. The 12 different colors in the figure represent the distinct signals recorded by each electrode. The location indicated by the red arrow is the pacing spike, which is a small electrical signals generated by a catheter to stimulate the ventricle. The onset, marked by a white point in Figure 2, signals the start of electrical activity that leads to the QRS complex, highlighted in the green box, which represents ventricular contraction. The QT interval Goldenberg et al. (2006), shown in the blue box, covers the electrical activity of both ventricular contraction and relaxation.

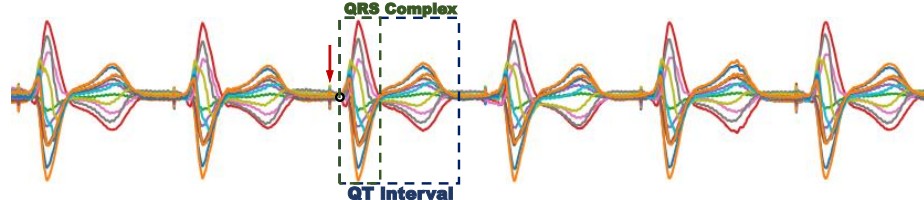

Figure 2: A segment of a 12-lead pacing ECG signals. The red arrow indicates the pacing spike, which are small electrical signals generated by a catheter to stimulate the heart. The green box highlights the QRS complex, the blue window shows the QT interval, and the onset, marked by a white point, represents the start of both.

## 3.2 PACING-SITE LOCATIONS

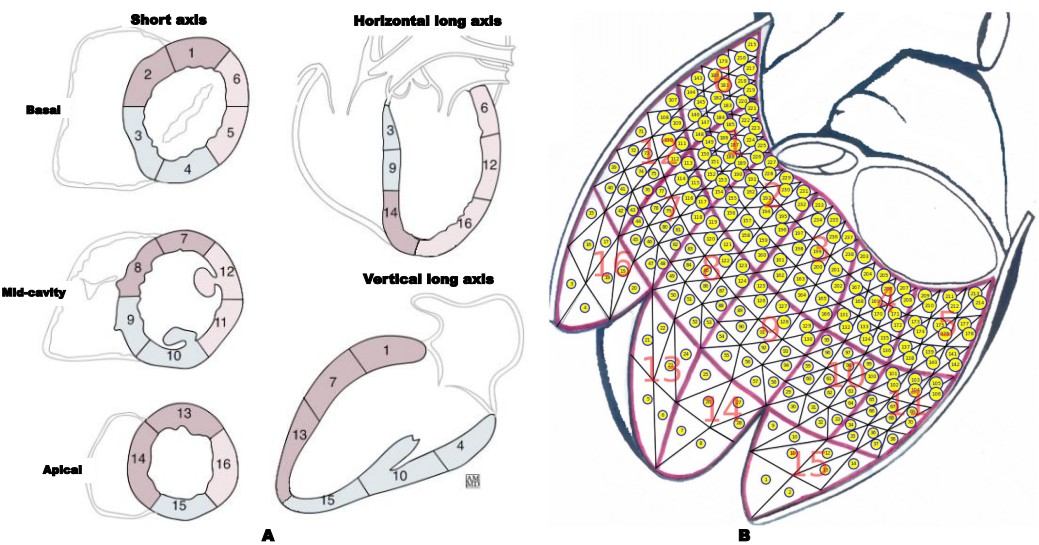

Figure 3: Two different methods for localizing the SoO. The first method divides the LV endocardial surface into 16 regions, as indicated by figure A. The second method, as shown in figure B, localize the SoO into 238 regular triangles, each marked by a yellow circle with a number.

The 3D geometry of the generic LV endocardial surface, derived from a necropsy specimen of a normal human heart Sapp et al. (2017), was divided into 16 anatomical segments (indicated in Figure 3A). These segments were further divided into 238 triangular elements to represent the LV endocardium Sapp et al. (2017) (as shown in the Figure 3B). Each triangle is defined by three Cartesian coordinates (X, Y, and Z), with the origin at the LV apex and the Z-axis oriented toward the midpoint of the mitral valve. For clinical application, pacing sites from patient-specific electroanatomic mapping data were manually registered onto these triangular elements within the 16 segments.

## 4 SIMBOL FRAMEWORK

### 4.1 ONSET-BASED DATA AUGMENTATION

In the preprocessing of pacing ECG data, not all pacing beats are clinically usable due to factors such as motion artifacts, ectopic beats (highlighted by the red dashed box in Figure 4), and non-captured beats where the stimulus-QRS delay exceeds 40 ms. Previous studies, such as Zhou et al. (2020a), used a single beat per pacing site and represented each lead's ECG using a 120 ms QRS integral to identify the SoO of early LV activation. While this approach showed strong localization ability Zhou et al. (2020a), it may miss important information beyond the initial 120ms of the QRS complex, limiting usable training data and introducing subjectivity in determining the optimal 120-ms QRS interval. To overcome these limitations, SimBOL produced Onset-based Data Augmentation strategy. This strategy aims to increase the availability of training data by generating additional

samples and simplifies the selection of QRS onset regions. By enhancing the dataset size and reducing the complexity of data preprocessing, SimBOL improves the robustness of ECG analysis and facilitates more accurate clinical assessments.

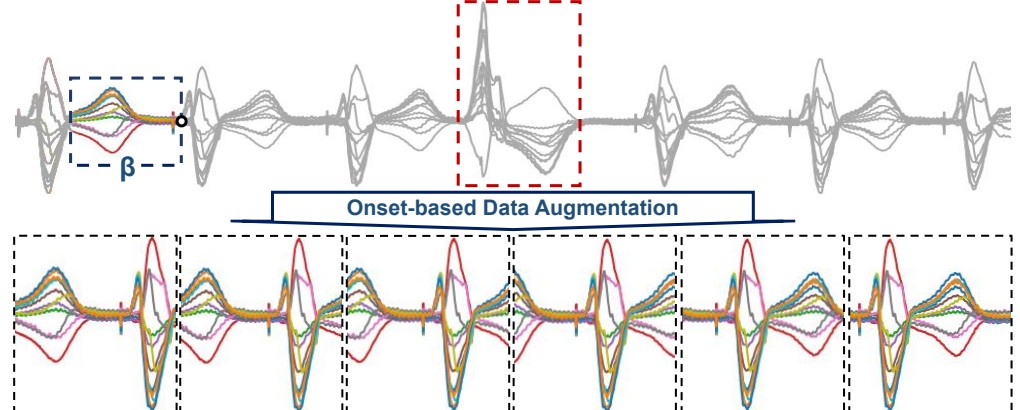

Figure 4: The onset-based data augmentation strategy and some of the resampling results. The blue dashed box indicates the resampling interval $\beta$. The white dot represents the optimal onset point. The red dashed box indicates the unusable, irregular QT interval.

Assuming the sampling rate of a sample is $F$, the duration of the QT interval is less than $\Delta$, and time interval between two spikes is $\Gamma$, the actual number of sample points $P$ within a QT interval is $P = F \times \Delta$ and the actual number of sample points $L$ between two spikes is $L = F \times \Gamma$. To determine the optimal onset time, which corresponds to point $t$, we define the interval $\beta = [\, t\text{-}\frac{P}{2},\, t \,]$ as the range for selecting the resampling start point. SimBOL augments the training dataset by randomly choosing the start point $\tau$ from this $\beta$ interval and then sampling $L$ points from $\tau$, forming a single training data sample. As shown in Figure 4, since $L > P$, each sample will include a portion or the entirety of the QT interval. Any portion of $P$ that is not captured within the sample is filled by adjacent QT interval data. By performing $N$ resamplings at each onset point, SimBOL$_{\times N}$ generates a training dataset that is $N$-fold larger than the original training dataset.

## 4.2 MODEL ARCHITECTURE

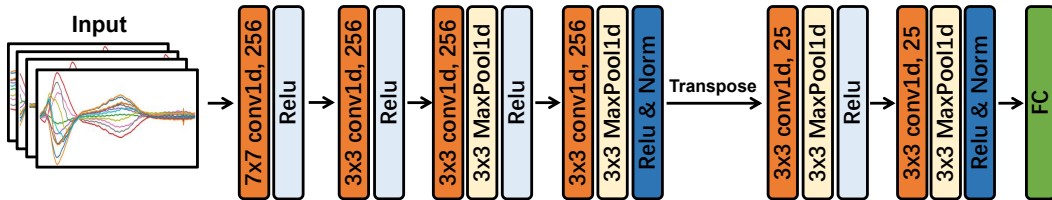

Figure 5: Illustration of the specific structure of the model used by SimBOL. It mainly consists of two stacked 1D convolutional blocks, followed by a fully connected layer.

By leveraging the physical characteristics of ECG signals, we developed a compact training model that combines 1D convolution with activation functions. This model has a limited number of parameters, helping to balance the scale of training data and model complexity, while effectively maintaining the ability to extract features from ECG signals. The detailed model architecture is illustrated in Figure 5. Based on actual clinical ECGs, SimBOL sets the input length of the resampled signals, denoted as $L$ (detailed in Section 4.1), to 800. Consequently, the model first employs a $7 \times 7$ 1D convolution to extract coarse-grained features from the data, followed by a $3 \times 3$ 1D convolution to capture finer-grained features. The most significant single-sample temporal features within each sample are then extracted using activation functions and pooling layers. To ensure that the model captures both the temporal features of the ECG signals and the relationships among the 12 leads, SimBOL transposes the obtained data representation and then applies a $3 \times 3$ 1D convolution to extract the sequential relationships between the leads. The final representation, $F$, is a

1600-dimensional vector. Finally, a fully connected layer is used to generate the coordinates of VA origin corresponding to the input sample.

### 4.3 EVALUATION PROTOCOL

SimBOL employs a coordinate prediction strategy for localization. Specifically, the model's final fully connected layer is designed to output a 3-dimensional coordinate $P = (x, y, z)$. The training loss $L$, shown in equation 1, is then calculated by measuring the Euclidean distance between the predicted coordinate and the reference coordinate $P' = (x', y', z')$.

$$L = ||P - P'||^2 \tag{1}$$

## 5 EXPERIMENTS

### 5.1 EXPERIMENT DATASETS

In this study, we utilized a pacing-site dataset containing 1,012 LV pacing sites, as referenced in Sapp et al. (2017). Each pacing site includes paired data: electrocardiograms (ECGs) and their corresponding pacing locations. The ECG recordings consist of approximately 15 seconds of continuous 12-lead pacing signals, captured at a sampling rate of 1,000 Hz during routine pace-mapping procedures on the LV endocardium. The pacing locations were mapped onto the generic LV composed of 238 triangular segments. Each site is defined by the $(x, y, z)$ coordinates of the center of one of these triangles. Notably, the 1,012 pacing sites do not cover all 238 triangle labels in the model. On average, each triangle is associated with fewer than 10 clinical samples, and some triangles have as few as one or even no samples. Specifically, 25 triangle labels have no associated clinical samples, and 27 labels have only one corresponding clinical sample. Details of the dataset are provided in the Appendix A.1.

### 5.2 EXPERIMENTS SETTING

#### 5.2.1 DATASET DIVISION STRATEGY

To ensure a reasonable division between the training and test sets, for triangle labels with more than two samples, we randomly selected 20% of the samples (rounded up) as the test set, with the remaining 80% used as the training set. For triangle labels with only two samples, we randomly selected one as the training sample and the other as the test sample. For triangle labels with only one sample, to increase the amount of training data, we included these samples exclusively in the training dataset. This approach resulted in 231 test samples and 781 training samples. SimBOL resamples each training sample using onset-based data augmentation.

#### 5.2.2 DATA AUGMENTATION WITH SIMBOL

SimBOL applies onset-based data augmentation by resampling each training sample multiple times. The number of resample $N$ for each training sample is determined by a resampling rate, $\times N$. Additionally, SimBOL resampled each test sample to enhance the generalization capability of the test set. In all experiments, we fixed the resamples rate for the test set at $\times 10$, resulting in 2,3,10 test samples for evaluation. We did not apply onset-based data augmentation to the original 1,012 clinical samples before splitting them into training and test sets. Doing so would have made the domains of the training and test sets identical, which would prevent the test results from accurately reflecting the model's generalization and potential overfitting.

#### 5.2.3 TRAINING SETTING

During the training process, SimBOL consistently set the training epochs to 400 and the batch size to 350. We used the Adam optimized with hyperparameters: beta1 =0.9, beta2 = 0.98, epsilon = $1e^{-9}$, and a weight decay of $1e^{-3}$. The model was trained using a cosine annealing learning rate schedule with 500 warmup steps and an initial learning rate of $1e^{-3}$. All experiments were conducted on 4 NVIDIA 4090 GPUs. To ensure the stability and reliability of the experimental

results, each experiment was run five times with different random seeds, and the mean and variance were calculated.

## 5.3 PERFORMANCE OF SIMBOL

### 5.3.1 COORDINATE PREDICTION ACCURACY

To compare the localization accuracy of different methods, we measured the average distance between the predicted coordinates and the corresponding reference coordinates on the test samples. The mean and variance of localization accuracy for each method are shown in Figure 6. To compare the performance differences between SimBOL and other methods, Figure 6 shows the different performance of the SimBOL under different resampling rates: ×1 (SimBOL$_{\times 1}$), ×2 (SimBOL$_{\times 2}$), ×3 (SimBOL$_{\times 3}$), ×4 (SimBOL$_{\times 4}$), and ×5 (SimBOL$_{\times 5}$), as described in section 4.1. The value in parentheses in the legend correspond to the mean of localilzation accuracy for each method.

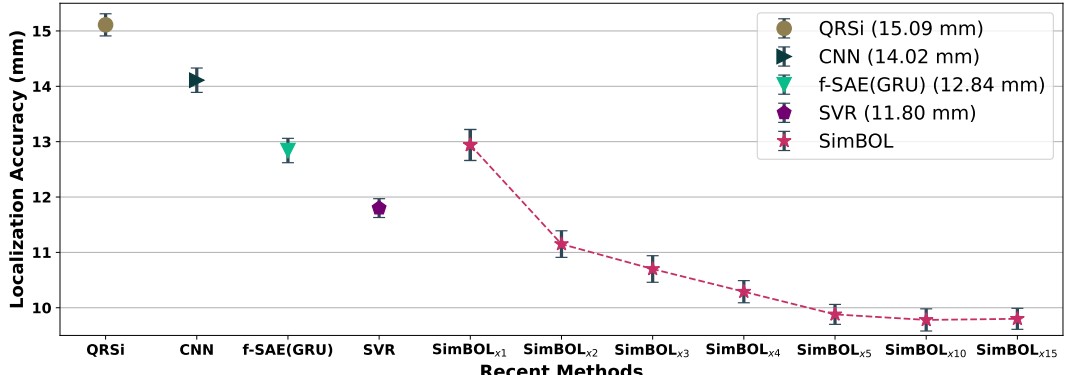

Figure 6: Comparison of the localization accuracy for each methods. To better illustrate SimBOL's performance, the figure shows the results at different resampling rates SimBOL$_{\times N}$. Each point in the figure represents the mean and variance of the average distance between the predicted coordinates and the corresponding true coordinates for the respective method across all test sets.

The QRS integral (QRSi) Sapp et al. (2012), used as a baseline in VA origin localization, is a linear model that employs the commonly-used feature of 120-ms QRS integrals, calculated as the 120-ms time integral of the QRS complex using the standard trapezoidal rule. The CNN model Yang et al. (2017) is inspired by premature ventricular contraction (PVC) localization from 12-lead ECG data and uses 2D convolution layers to extract features. The results in the Figure 6 also indicate that using 2D convolution to jointly extract temporal features and 12-lead relationships from ECG signals does not demonstrate a significant advantage. The f-SAE(GRU) Gyawali et al. (2019) is a deep model based on GRU modules Dey & Salem (2017), where the GRU is pre-trained through a generative task to mitigate overfitting during the training phase of the SoO localization. Although this method effectively improves the accuracy of predicting the SoO, the overall training process is complex, the model has a large number of parameters, and it requires specific input intervals for the data. The SVR model Zhou et al. (2019) is a linear regression model based on 120-ms QRS-integrals. It achieved an average localization accuracy of 11.80 mm, making it the most accurate method to date. However, this method is highly sensitive to the selection of the QRS onset; accurately identifying the QRS onset in ventricular arrhythmias is a challenging task due to the irregular and variable nature of arrhythmic ECG signals Reznichenko S & S (2024).

By analyzing the results of SimBOL, we observe that its localization accuracy improves as the resampling rate increases. With only one-fold resampling, SimBOL achieves a localization accuracy of 12.94 mm, which is very close to that of f-SAE(GRU) at 12.84 mm, and significantly higher (worse performance) than SVR. However, as the resampling rate increases, SimBOL's localization error decrease from 12.94 mm to 9.88 mm at a ×5 resampling rate. However, as the amount of training data continues to increase, the model's performance did not show significant improvement and stabilized around 9.80 ± 0.19 mm. This result aligns with the empirical conclusion Krizhevsky

et al. (2017); Chen et al. (2020); Ni et al. (2023); Dubey et al. (2024) in traditional deep learning: model performance improves with an increase in training data and eventually reaches a plateau.

### 5.3.2 THE PERFORMANCE IN 16 SEGMENTS

This section evaluates SimBOL's localization accuracy across all test samples by classifying the results into 16 segments, showing how the model's accuracy varies across the 16 segments at resampling rates: $\times 1$, $\times 3$, $\times 5$, $\times 10$ and $\times 15$. Detailed comparison results are presented in Figure 7. The results in parentheses in the legend represent the average coordinate prediction distance for the SimBOL model at each resampling rate, which is consistent with the results in Figure 6.

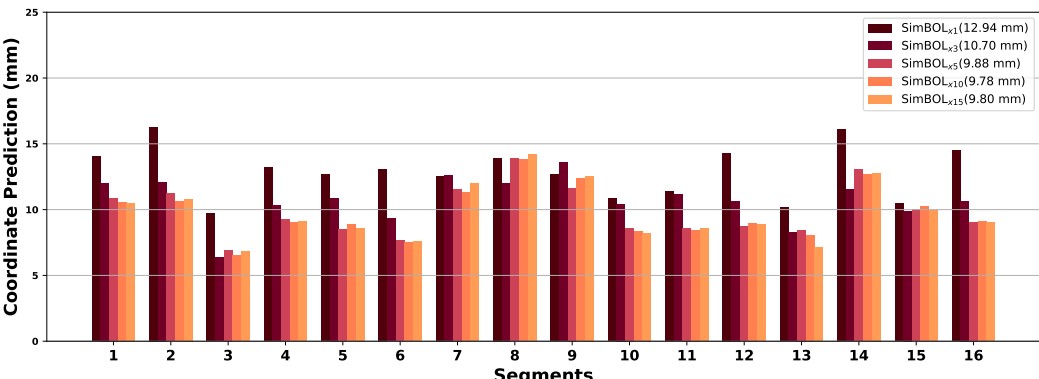

Figure 7: Coordinate prediction distance of SimBOL models across 16 segments at five different resampling rates: $\times 1$, $\times 3$, $\times 5$, $\times 10$, and $\times 15$. The results in parentheses in the legend represent the average coordinate prediction distance for the SimBOL model at each resampling rate.

Comparing the results of SimBOL models with five different resampling rates reveals that, in most segments, the localization error decreases as the resampling rate (and thus the amount of training data) increases. The most significant performance improvements are observed between the $\times 1$, $\times 3$, and $\times 5$ resampling rates. Beyond $\times 5$, the differences between SimBOL$_{\times 5}$, SimBOL$_{\times 10}$, and SimBOL$_{\times 15}$ are not significant in most segments, indicating a plateau in performance gains.

However, localization results for four segments, specifically segment 7, 8, 9, and 14, are notably unstable. In these segments, the localization accuracy did not improve or remain stable with the increase in training data; instead, it actually worsened. As shown in Figure 3 A, segments 8, 9 and 14 correspond to the ventricular septum, an area of the heart with complex anatomical features. The presence of the conduction system, including Purkinje fibers, makes electrical signals from these regions difficult to localize. Moreover, fewer pacing data were collected from these segments, which may have contributed to the decreased accuracy. For segment 7, we also observed poor localization results. This could be because segment 7 is located near the papillary muscles, where many pacing points are not on the endocardial surface but within the intraventricular area. The unique anatomical features of this region make it challenging for the model to accurately localize pacing sites. These findings not only demonstrate the overall stability of the SimBOL model but also highlight key areas—specifically segments 7, 8, 9, and 14—that significantly impact the model's performance. Identifying these challenging regions provides direction for future improvements in finer-grained localization techniques.

### 5.4 THE INFLUENCE OF DATA AUGMENTATION

To analyze the impact of different data augmentation strategies on the performance of the SimBOL model, we incorporated several traditional data augmentation methods—Noise Augmentation (NA), Amplitude Scaling Augmentation (ASA), and Random Baseline Wander Augmentation (RBWA)—on top of the Onset-based Data Augmentation (ODA). As shown in Figure 8, The NA adds random noise within a specific range $R$ to each point of the resampled input. The ASA scales the original input sample by multiplying it with a random factor within a specific range $Q$. The RBWA introduces a sinusoidal wave to the original input, with the wave's period matching the sample length, and its amplitude drawn from a specific range $Z$. Details of the augmentation settings

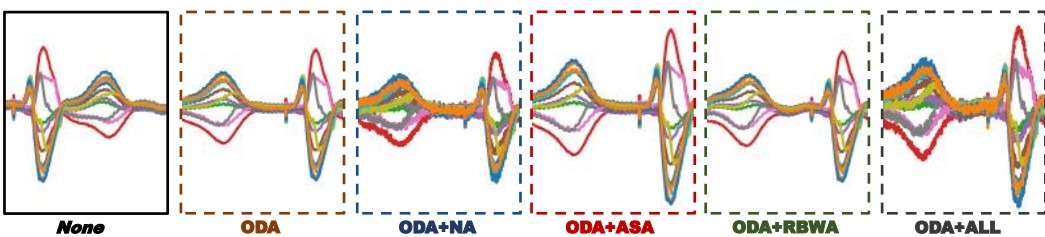

Figure 8: Illustration of the training data without data augmentation ($None$) and with five different data augmentation combinations. ODA refers to Onset-based Data Augmentation, NA stands for Noise Augmentation, ASA represents Amplitude Scaling Augmentation, RBWA refers to Random Baseline Wander Augmentation ,and ALL represents the combination of NA, ASA, and RBWA.

can be found in the Appendix A.5. For this experiment, $R$, $Q$, and $Z$ were all within the range $[0.15, 0.25]$. Table 1 illustrates the effect of applying various combinations of data augmentations at different resampling rates $\times N$ on the coordinate prediction distance of the SimBOL model. The column marked $None$ refers to the case where no data augmentation, including ODA, was applied. These results are denoted by '/', as there were no resampling or data augmentation processes involved. The bold values represent the lowest prediction distance error results for the model at the current sampling rate.

Table 1: Comparison of the impact of different data augmentation combinations and resampling rates on the performance of the SimBOL model.

| $\times N$ | $\times 1$ | $\times 2$ | $\times 3$ | $\times 4$ | $\times 5$ | $\times 10$ | $\times 15$ |
|---|---|---|---|---|---|---|---|
| ODA | 12.94±0.28 | **11.15±0.24** | 10.70±0.24 | 10.29±0.20 | 9.88±0.18 | **9.78±0.20** | 9.80±0.19 |
| +NA | 13.41±0.32 | 11.65±0.30 | 11.23±0.31 | 10.88±0.26 | 10.12±0.23 | 10.12±0.21 | 10.14±0.23 |
| +ASA | **12.81±0.26** | 11.18±0.24 | **10.68±0.22** | 10.33±0.21 | 9.84±0.20 | 9.80±0.18 | 9.92±0.20 |
| +RBWA | 13.64±0.33 | 12.21±0.26 | 11.10±0.27 | 10.34±0.23 | 10.02±0.22 | 9.82±0.21 | **9.76±0.23** |
| +ALL | 13.81±0.25 | 11.75±0.22 | 10.92±0.23 | **10.25±0.22** | **9.83±0.20** | 9.80±0.18 | 9.88±0.19 |
| $None$ | 12.95±0.22 | / | / | / | / | / | / |

From the results in the table, it is clear that adding Noise Augmentation (+NA) on top of ODA has the most significant negative impact on the model's performance. Regardless of the resampling rate, the model's prediction accuracy consistently decreased after introducing NA. We believe this is due to the unstructured noise disrupting the temporal features of the input data, making the relationship between ECG features and their corresponding coordinates less stable, which in turn affect the model's predictions. In contrast, both ASA and RBWA slightly improved the prediction accuracy of the SimBOL model at certain sampling rates compared to using only ODA. However, these methods introduced a slight decrease in stability, as indicated by the increased variance in results. When comparing the results of ODA alone with ODA combined with all three traditional data augmentation methods (+ALL), it becomes evident that after increasing the training data (when the resampling rate reached $\times 4$), the model's performance shows no significant difference from using ODA alone. Additionally, comparing the model's performance with ODA to the "None" case (no data augmentation), it is apparent that SimBOL's model is not highly sensitive to input data structure, as the final average prediction error is the same in both cases. However, once the training data increases through resampling, the importance of ODA becomes clear. In every combination of data augmentation strategies, the model's performance consistently improves as the amount of training data increases.

## 5.5 THE INFLUENCE OF MODEL PARAMETERS AND STRUCTURE

Extensive research in deep learning have demonstrated the strong feature extraction capabilities of the Transformer architecture Vaswani (2017) particularly for sequential signals. In this section, we

discuss the impact of inserting multi-head self-attention layers Niu et al. (2021) at different layers of the SimBOL model on its final coordinate prediction accuracy. Specifically, we experimented with adding two stacked multi-head self-attention layers (denoted as T) either before the input layer (T+SimBOL) or before the fully connected layer (SimBOL+T) in the SimBOL model. For detailed information on the structure of T, please refer to Appendix A.6. Figure 9 shows the average coordinate prediction performance of these models under different resampling rates $\times N$. The values in parentheses in the legend represent the average prediction error for each model when the resampling rate reaches $\times 100$.

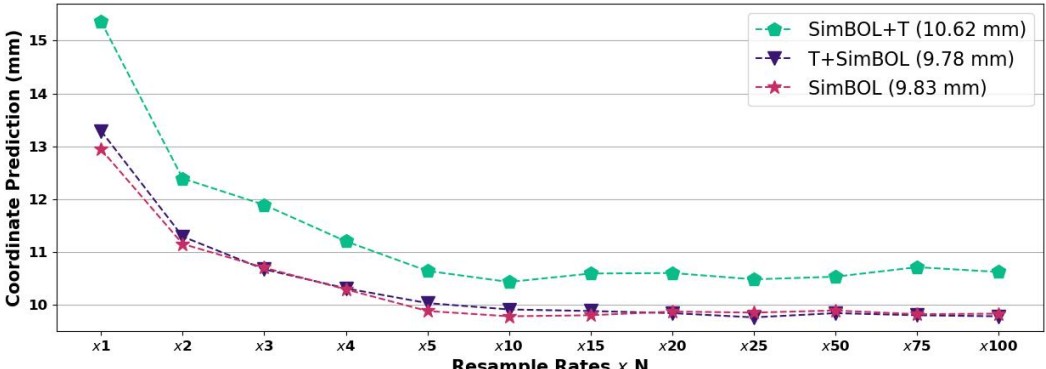

Figure 9: The trend in coordinate prediction accuracy at different resampling rates for three different model architectures: SimBOL+T, T+SimBOL, and SimBOL). The results in parentheses in the legend represent the average coordinate prediction distance for these model at resampling rate $\times 100$.

The results in the figure 9 show that adding a Transformer structure before the fully connected layer (SimBOL+T) leads to a decreases in performance. As the resampling rate increases, the prediction error for the SimBOL+T model decreases from 15.36mm at the $\times 1$ to 10.62mm at the $\times 100$. However, this is still 0.79mm higher than the original SimBOL model. We believe this is because, after passing through the SimBOL module, the data features become highly abstracted, making the T structure less effective. Additionally, the large number of training parameters in the T likely hinders overall model training. In contrast, the T+SimBOL model, which applies the self-attention structure to create a data representation with consistent dimensions, preserve a significant amount of detailed information of the data, and then processes this representation with the SimBOL module for high-dimensional mapping. The T+SimBOL model achieves a prediction accuracy that is similar to the original SimBOL model, which only a 0.05mm difference at the $\times 100$ resampling rate. Comparing the accuracy curves, we observe that the T+SimBOL model stabilizes at a resampling rate of $\times 15$, whereas the SimBOL model stabilizes at $\times 5$ due to its smaller number of parameters.

The performance trends of the T+SimBOL and SimBOL models illustrate that a model's effectiveness is influenced by the balance between the number of model parameters and the amount of training data. Larger model scales, such as T+SimBOL, require more training data to support their performance. The comparison between the T+SimBOL and SimBOL+T models demonstrates that the model structure has a significant impact on the final performance.

## 6 CONCLUSION

This paper presents a **Sim**ple data-parameters **B**alancing framework for ventricular arrhythmia **O**rigin **L**ocalization (SimBOL) in ECG filed. The proposed onset-based data augmentation method offers a new strategy for expanding training data in the ECG domain for deep learning. The produced small-scale model, built using stacked 1D convolutions, ensures effective feature extraction from ECG data while reducing the preprocessing demands of the signals. By balancing training data and model parameters, SimBOL achieves a localization accuracy of under 10 mm on the unified test set, which meets clinically-accepted accuracy and improved accuracy by 2 mm compared to the current best method, SVR. The experimental discussion demonstrates the performance stability of the SimBOL model and emphasizes the importance of balancing training data and model parameters. The discussion about data augmentation and model architecture on ECG signal processing, offering new insights into optimizing deep learning applications for ECG-based tasks.

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

# A  APPENDIX

## A.1  DETAILED INTRODUCTION OF EXPERIMENTAL DATA

The 1,012 distinct clinical samples as experimental data, collected from 39 patients who underwent ablation for scar-related VT. The study protocols of this dataset were approved by the Nova Scotia Health Authority Research Ethics Board Hub. The specific 25 triangle labels with no corresponding clinical samples are: 35, 42, 52, 78, 125, 127, 132, 152, 158, 159, 160, 161, 164, 191, 194, 196, 198, 199, 200, 201, 217, 225, 229, 233, 237. The specific 27 triangle labels have only one corresponding clinical sample are: 40, 47, 54, 56, 90, 91, 117, 120, 126, 129, 141, 147, 162, 171, 176, 192, 193, 195, 197, 210, 213, 216, 228, 231, 232, 234, 235. Figure 10 shown the number of clinical training and testing data for each triangle label.

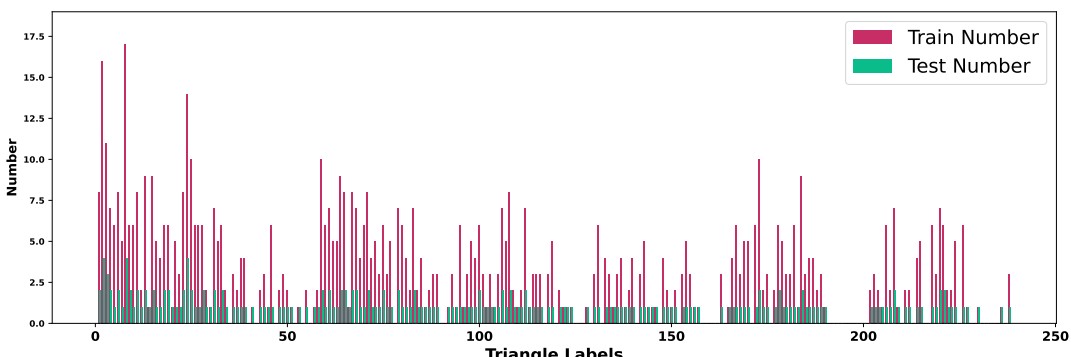

Figure 10: The number of clinical training and testing data for each triangle label.

## A.2  DETAILED COORDINATE PREDICTION RESULTS

Section 5.3.1 presents the average coordinate prediction accuracy of the SimBOL model. Here, we show the detailed localization accuracy of SimBOL$_{\times 5}$ for each triangle label in Figure 11.

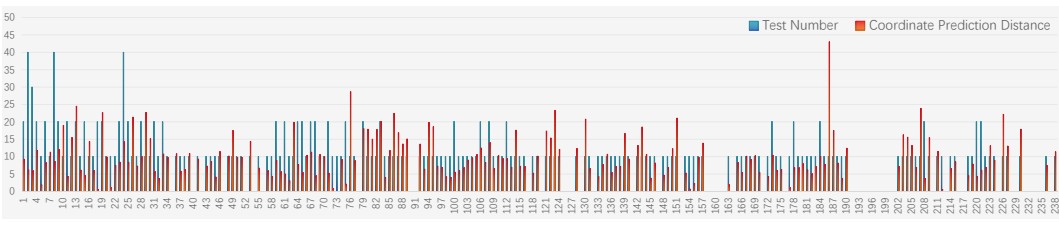

Figure 11: The detailed localization accuracy of SimBOL$_{\times 5}$ for each triangle label

From the overall results, it can be observed that SimBOL's localization accuracy shows little variation across most triangle labels. However, there are noticeable fluctuations in the accuracy for several triangle labels, such as 13, 76, 123, 186, 207, and 226. In particular, the localization error for triangle 186 reached 42.97mm, making it an outlier in the model's performance. A total of 12 triangular coordinates had a prediction distance exceeding 20mm. Among these, 8 triangle labels were located in segments 1, 7, 8, 9, and 14, which correspond to the ventricular septum region (section 5.3.2). The least accurate, triangle 186, was found in segment 1.

## A.3 PERFORMANCE OF 16 SEGMENTS CLASSIFICATION

Besides discussing SimBOL's performance in coordinate prediction, this section reports the $\text{SimBOL}_{\times 5}$ average classification accuracy across all test samples for the 16 predefined segments. The final results are shown in Figure 12.

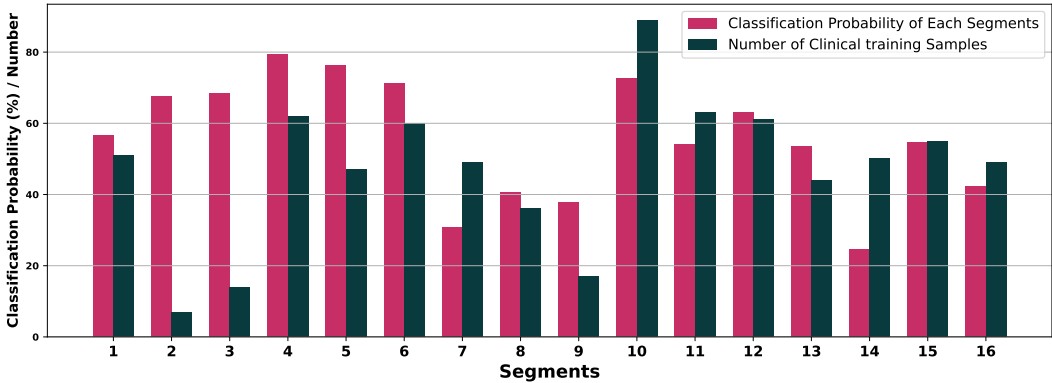

Figure 12: Illastration the classification accuracy of the $\text{SimBOL}_{\times 5}$ on 16 segments.

While the segmentation into 16 regions aids physicians in identifying the broader area of interest, approximately pinpointing the SoO of early ventricular activation within these regions often necessitates extensive pacing. This is because discrete localization can create large target areas and potentially overlook adjacent segments, particularly when the SoO lies on a boundary shared by two or three neighboring segments. The results in the figure show that the classification accuracy of segments 7, 8, 9, and 14 is significantly lower than that of other segments. This finding aligns with the results discussed in section 5.3.2. In conjunction with the results from Section 5.3.2, it is evident that the $\text{SimBOL}_{\times 5}$ model's predicted coordinates in segments 7, 8, 9, and 14 are not only far from the reference coordinates but also located in different segments.

## A.4 THE INFLUENCE OF $\beta$ INTERVAL

In the manuscript (section 4.1), we set the resampling interval for onset-based data augmentation as $\beta = [\, t\text{-}\frac{P}{2}, \text{t} \,]$, where the interval length is $P/2$. In this section, we expanded the resampling interval into $\beta = [\, t\text{-}\frac{P}{2}, \text{t} + \frac{P}{2} \,]$, thereby increasing the interval length to $P$. Table 2 presents the performance of the SimBOL model at different resampling rates under two different resampling intervals. The column marked "None" refers to the case where no data augmentation was applied. These results are denoted by '/', as there were no resampling or data augmentation processes involved. From the results, we can see that expanding the resampling interval did not lead to a significant difference in the model's final localization performance (the performance in $\times 15$, $\times 20$ and $\times 25$). As the resampling interval expanded, the diversity of the training data increased, which resulted in a slight increase in the variance of the model's localization accuracy. However, at higher resampling rates, the increased training data contributed to more stable model performance.

Table 2: Comparison of the impact of different $\beta$ interval and resampling rates on the performance of the SimBOL model.

| $\times N$ | $\times 1$ | $\times 3$ | $\times 5$ | $\times 10$ | $\times 15$ | $\times 20$ | $\times 25$ |
|---|---|---|---|---|---|---|---|
| $P/2$ | 12.94±0.28 | 10.70±0.24 | 9.88±0.18 | 9.78±0.20 | 9.80±0.19 | 9.87±0.19 | 9.85±0.19 |
| $P$ | 13.01±0.35 | 12.17±0.30 | 11.88±0.25 | 10.12±0.22 | 9.90±0.23 | 9.85±0.21 | 9.91±0.23 |
| $None$ | 12.95±0.22 | / | / | / | / | / | / |

## A.5 DETAILED SETTINGS FOR DATA AUGMENTATION

Section 5.4 compares the impact of different data augmentation combinations on the performance of the SimBOL model. Here, we provide a detailed algorithm of the implementation details for the three data augmentation methods.

---

**Algorithm 1** Noise Augmentation

---
import $torch$; import $numpy$ as np
**input:** 12-lead ECG signals $X$, noise_ranges $R$.
noise = np.random.uniform($R$).
**for** i in range($X$.shape[0]): **do**
    $X$[i] = $X$[i] + noise * np.random.randn($X$[i].shape[0])
**end for**
**return** $X$

---

**Algorithm 2** Amplitude Scaling Augmentation

---
import $torch$; import $numpy$ as np
**input:** 12-lead ECG signals $X$, factor_ranges $Q$.
factor = np.random.uniform($Q$).
$X = X$ * factor
**return** $X$

---

**Algorithm 3** Random Baseline Wander Augmentation

---
import $torch$; import $numpy$ as np
**input:** 12-lead ECG signals $X$, wander_ranges $Z$.
wander = np.random.uniform($Z$).
t = $torch.linspace$(0, 1, $X$.shape[1])
t = t.unsqueeze(0)
t = t.repeat($X$.shape[0],1)
$X = X$ + wander*$torch.sin$(2*$torch.pi$*t)
**return** $X$

---

## A.6 DETAILED SETTINGS FOR TRANSFORMER

Section 5.5 discusses the impact of model architecture and parameters on the overall performance of the SimBOL model. Table 3 provides a detailed explanation of the T structure in both the T+SimBOL and SimBOL+T models. To minimize changes to the model, we only adjusted their input dimensions (dim model), while keeping all other training parameters identical to those used in SimBOL (detailed settings can be reviewed in Section 5.2). the Figure 13 show the detailed framework of T+SimBOL. And the Figure 14 show the detailed framework of SimBOL+T.

| Hyperparameter of T+SimBOL | Value | Hyperparameter of SimBOL+T | Value |
|---|---|---|---|
| num blocks | 2 | num blocks | 2 |
| dim model | 800 | dim model | 1600 |
| ff ratio | 4 | ff ratio | 4 |
| num heads | 4 | num heads | 4 |
| kernel size | 3 | kernel size | 3 |
| Pdrop | 0.1 | Pdrop | 0.1 |

Table 3: A detailed explanation of the T structure (in Section 5.5) within both the T+SimBOL and SimBOL+T models.

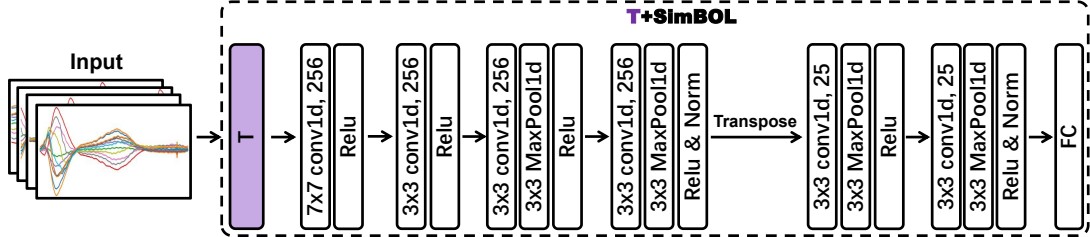

Figure 13: The detailed framework of SimBOL+T model.

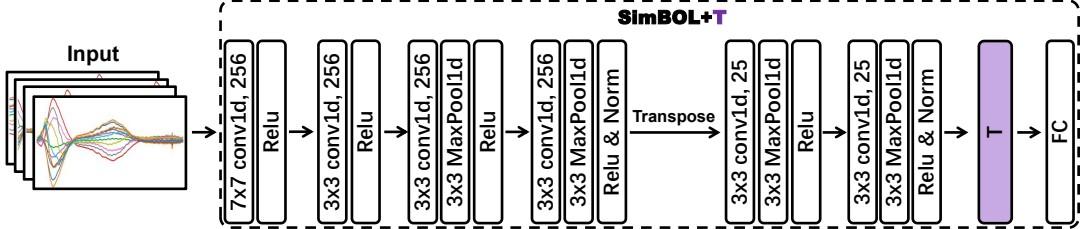

Figure 14: The detailed framework of SimBOL+T model.

