# OpenReview forum: "A Simple Data-Parameters Balancing Framework for Early Ventricular Activation Origin Localization"
_ICLR.cc/2025/Conference — ICLR 2025 Conference Withdrawn Submission_

### Official Review · Reviewer_hQo9 · 2024-10-23

**Soundness:** 3
**Presentation:** 3
**Contribution:** 2
**Rating:** 3
**Confidence:** 5

**Summary:**

This paper presents a new method for localizing the origin of early ventricular activation from 12-lead ECG. The main contribution is an onset-based data augmentation, and a small scale 1D convolution model.  Experimental evaluation is conducted on a private dataset. The performance of the model is evaluated against a SVR model and two deep learning model in the Euclidean distance of localization.

**Strengths:**

- The presented data augmentation strategy demonstrated an ability to improve the accuracy in localizing early activation when training data is limited.
- The presented model demonstrated comparable and improved performance to the included baselines with a small-scale model.

**Weaknesses:**

1. The paper largely motivate the presented method by stating that the performance of existing models is significantly constrained by overfitting — there is however no obvious evidence substantiating this statement. While the performance of the baseline deep learning models included showed inferior performance to the presented method, it is not clear if they are overfitting.
2. Given the large number of exiting baselines, including those reviewed in the paper, a much larger number of baselines should be considered in the experiment. In addition, model-based approaches that are utilizing simulation models instead of data-driven models should also be considered as baselines.
3. The label of the xyz coordinates appears to be limited to the resolution of the triangle (i.e., it seems that there are only 238 unique xyz labels), instead of true x,y, z coordinates. This goes against the stated contribution #3.
4. Without augmentation it seems that the presented model was performing similar to one of the baselines and worse than SVR — it is thus not clear if the gain in performance mainly comes from the augmentation strategies rather than the model architecture. It could be interesting to test baseline methods with the same augmentation methods (since it is not tied to the particular architecture), to demonstrate the more general benefits of the proposed augmentation method.
5. What is a 7x7 (or 3x3) 1D convolution?
6. Overall, while an interesting application, the presented methodology is of limited technical innovation and may be better suited for an application-oriented venue instead of ICLR.

**Questions:**

1. Please provide specific evidence or analysis demonstrating that existing models are overfitting, rather than just underperforming.

2. Please consider adding some of the following baselines to comparison:
[1] Monaci et al, 2023, Non-invasive localization of post-infarct ventricular tachycardia exit sites to guide ablation planning: a computational deep learning platform utilizing the 12-lead electrocardiogram and intracardiac electrograms from implanted devices
[2] Monaci et al 2020, In-silico pace-mapping using a detailed whole torso model and implanted electronic device electrograms for more efficient ablation planning
[3] He et al 2020, Localization of origins of premature ventricular contraction in the whole ventricle based on machine learning and automatic beat recognition from 12-lead ECG

3. Please clarify the resolution of the coordinate labels and how this relates to the claimed contribution of coordinate-based localization. Please discuss any limitations associated with the labels.

4. Please apply the onset-based data augmentation to the baseline methods and report the results, to isolate the impact of the augmentation versus the model architecture.

5. Please  provide more details on the convolutional layer implementation, specifically clarifying how 7x7 or 3x3 kernels are applied in a 1D convolution operation.

---

### Official Review · Reviewer_e2sk · 2024-11-04

**Soundness:** 2
**Presentation:** 2
**Contribution:** 2
**Rating:** 3
**Confidence:** 4

**Summary:**

This paper presents a CNN-based model for ventricular activation site localization, specifically estimating the origin location in Cartesian coordinates based on an ECG sequence.

**Strengths:**

Localizing the site of origin has substantial clinical impact, potentially guiding treatment decisions for cardiac patients.

**Weaknesses:**

1. The paper's writing needs significant improvement to align with the machine learning community's standards. It is currently more clinically oriented, leaving many key concepts (e.g., QRS and QT intervals) unexplained. Additionally, the localization problem lacks a formal mathematical formulation. The loss function in Eq. 1 is defined only for a single point and does not account for the distribution of the entire input sample.

2. The evaluation of the proposed data augmentation approach (Table 1) is unconvincing. Classical data augmentation methods are layered sequentially rather than applied in parallel with the proposed approach, making it difficult to properly assess the proposed method's unique effectiveness.

3. The study does not adequately address overfitting concerns, despite mentioning it in the introduction. There are many established methods to tackle overfitting, such as regularization, model pruning, and cross-validation. However, these methods were not explored, weakening the justification for the proposed approach.

4. The proposed approach was evaluated on only one dataset, raising concerns about its reproducibility and generalizability.

**Questions:**

In Figure 6, are the other approaches trained with the same augmentation setup as the proposed method? Since the proposed augmentation is intended to be architecture-agnostic, evaluating it on multiple architectures would provide stronger evidence of its effectiveness.

---

### Official Review · Reviewer_MZuE · 2024-11-07

**Soundness:** 2
**Presentation:** 2
**Contribution:** 2
**Rating:** 3
**Confidence:** 4

**Summary:**

This paper presents a novel data augmentation approach for ECG applications focused on localizing ventricular activation origins. The proposed data augmentation and method improve localization performance compared to existing methods on the tested dataset.

**Strengths:**

1. The paper addresses the significant challenge of ventricular activation localization using real-world clinical datasets.
2. The proposed augmentation method demonstrates improvements over existing approaches.

**Weaknesses:**

1. Novelty: The paper’s claim of proposing a novel data feature extraction model is not well justified, as it primarily employs standard 1D convolutional networks. While the proposed data augmentation appears novel, it essentially resembles strategies similar to random padding of input signals.

2. Weaknesses in the Experimental Section:
2.1. Although ventricular activation localization is an important application and relevant datasets may be limited, this paper would benefit from additional experiments with benchmark datasets, such as MIT-BIH, to demonstrate the generalizability of the proposed augmentation. 2.2. The paper repeatedly claims that existing methods are large in size, but it lacks any comparative analysis of the size of previous methods (e.g., CNN, f-SAE with GRU) versus the proposed approach.
2.3. The explanation regarding the 16-segment model and its significance is missing. Additionally, it is unclear why the paper focuses exclusively on the performance of SimBOL without comparing it to existing methods.
2.4. Since the paper emphasizes novel augmentation, it would be valuable to analyze the effectiveness of this augmentation when applied to other methods (e.g., CNN, f-SAE with GRU).

3. Presentation: The quality of the figures needs improvement, and the paper contains grammatical errors (e.g., line 94) that impact readability.

**Questions:**

See weaknesses.

---

### Note · Authors · 2024-11-12

I have read and agree with the venue's withdrawal policy on behalf of myself and my co-authors.